# A Niemann–Pick Disease Type C2 with Severe Pulmonary Involvement and Limited Therapeutic Options: A Case Report

**DOI:** 10.3390/children9121811

**Published:** 2022-11-24

**Authors:** Abdullah Al-Shamrani, Khalid Al-Shamrani, Ahmed Bin Mahfoudh, Ahmed Sarar Mohamed, Sarar Mohamed

**Affiliations:** 1Department of Pediatrics, Prince Sultan Military Medical City (PSMMC), Riyadh 11159, Saudi Arabia; 2Al Faisal University, Riyadh 11159, Saudi Arabia; 3College of Medicine, Al Marifah University, Riyadh 11159, Saudi Arabia; 4Department of Pediatrics, National Hospital, Riyadh 11159, Saudi Arabia; 5College of Medicine, The National University, Khartoum 11115, Sudan

**Keywords:** Niemann–Pick, lipid storage disease, alveolar proteinosis, lipoid pneumonia

## Abstract

Niemann–Pick disease type C (NPC) is an autosomal recessive lipid storage disorder. There are two types, NPC1, which is the predominant form (95%), and the rare NPC2, which represents less than 5% of the reported cases. Niemann–Pick disease type C2 usually presents with respiratory symptoms, cholestasis, neurological impairment, and hepatosplenomegaly. Case report: Here, we report a 3-year-old boy who presented to our hospital with exacerbation of chronic lung disease requiring invasive ventilatory support. He was previously diagnosed with interstitial lung disease. His parents used to instill olive oil in his nose (a few drops in each nostril daily for several months) to treat frequent nasal bleeding. A detailed history revealed prolonged neonatal jaundice for four months, with hepatosplenomegaly. In his second year, generalized hypotonia and delayed psychomotor development were observed. Upon presentation to our institute, chest CT showed evidence of intraparenchymal fat; therefore, lipoid pneumonia and lipid storage disease were suspected. The bronchoalveolar lavage results suggested pulmonary alveolar proteinosis (PAP). Whole-exome sequencing (WES) revealed a class one homozygous pathogenic variant in the *NPC2* gene. Our patient faced a range of difficulties, including prolonged mechanical ventilation and diagnostic and therapeutic challenges. Conclusion: Niemann–Pick disease type C2 is a progressive and lethal condition that requires a high index of suspicion to pinpoint the diagnosis. Gene study remains the method of choice to confirm the diagnosis. There are limited choices of therapeutic interventions; therefore, genetic counseling and the prevention of recurrence should be the ultimate goal for affected families.

## 1. Introduction

Niemann–Pick disease (NPD) is a heterogeneous group of lysosomal storage disorders inherited in an autosomal recessive manner [1]. Niemann–Pick disease is classified based on its genetic and clinical features; there are four types, namely, type A, type B, type C1, and type C2. The affected gene in type A and type B is *SMPD1,* which causes an enzymatic deficiency of sphingomyelinase (SMA) [2]. Niemann–Pick disease type A usually presents with neurological manifestations leading to death in infancy. Niemann–Pick disease type B is the predominant type presenting with visceral enlargement and often surviving into adulthood. Niemann–Pick disease type C is a fatal condition due to mutations in either the *NPC1* or *NPC2* gene. Defects in the NPC1 gene account for approximately 95% of patients, but NPC2 is the cause in approximately 5% of reported cases [3,4,5].

Niemann–Pick disease is caused by the impaired trafficking of unesterified cholesterol and glycolipids in lysosomes and late endosomes, leading to the disturbed transport of various intracellular lipids, the reduced degradation of glucosylceramide, and the lysosomal storage of cholesterol and glycosphingolipids [6]. Lipid accumulation predominantly in the brain leads to functional and structural brain damage, visceral organ involvement (prolonged neonatal jaundice and hepatosplenomegaly), and pulmonary infiltrates [7,8,9,10]. This article highlights the diagnostic challenges, and the limited therapeutic options, for a boy with NPC2 disease and the impact of the disease on the family and future pregnancies.

## 2. Case Report

A 3-year-old Pakistani boy was admitted to our institute with an exacerbation of chronic lung disease. He was born at full term by spontaneous vaginal delivery with a weight of 3.0 kg, and there were no antenatal concerns. The parents are first cousins with no family history of infant death, developmental delay, or genetic disorder. The neonatal period was marked by jaundice, requiring a 4-day course of phototherapy. Later, in the neonatal period, he developed prolonged jaundice with cholestasis characterized by mixed hyperbilirubinemia, elevated liver enzymes, and hepatomegaly.

In the first year of life, he had frequent epistaxis, for which he used to receive olive oil in the nostrils, aiming to control the bleeding and nasal dryness. In the second year, the patient had a mild course of COVID-19 infection that did not require hospital admission. During his second year (a year ago), he also developed frequent respiratory symptoms that required admission to the ward on many occasions and responded to antibiotics, inhaled steroids, and bronchodilators. 

Six months ago, he presented with cough, dyspnoea, and desaturation, requiring admission to the pediatric intensive care unit (PICU) and ventilation for 2 days. Computerized tomography (CT) of the chest (Figure 1) showed diffuse homogenous ground glass opacity with interstitial septal thickening, which was interpreted by the primary physician as a picture of childhood interstitial lung disease (ChILD) and there was a subsequent prescription of steroids and hydroxychloroquine. He partially responded to treatment and was later discharged home on oxygen by nasal cannula. 

In his recent admission to our institute, at the age of three, he had another respiratory exacerbation requiring 5 L/m oxygen. On examination, he was in respiratory distress with subcostal and intercostal recessions, and his respiratory rate was 50 breaths per minute. His oxygen saturation was 70% in room air. He required 5–6 L/m oxygen through the nasal cannula to maintain saturation above 90%. He was febrile, had a heart rate of 140 beats per minute, and had blood pressure of 110/75 mmHg. His weight, length, and head circumference were between the 25th and 50th percentiles. He had grade three clubbing in both hands. Respiratory examination revealed markedly reduced breath sounds at the lung bases with fine inspiratory crackles and absent bronchial breath sounds. His splenic tip was palpable, and his liver was palpable at 4 cm below the costal margin. Neurological assessment revealed global developmental delay with central hypotonia and normal reflexes. Fundoscopy was normal.

## 3. Results

### 3.1. Chest X-ray 

There were homogenous ground glass opacities involving the entire right lung and the left upper and middle zone with mesocardium due to mild rotation to the right side; there was some retrocardiac collapse and secondary hyperinflation of the left lower lobe but there was no attenuation of the blood vessels to suspect pulmonary emphysema as a possible sign of NPC2 (Figure 2). 

### 3.2. Blood Work

The blood work, including normal complete blood count, liver, and renal function, and coagulation profile, was benign. On 2 L nasal cannula oxygen, the venous blood gas, consistent with type 2 respiratory failure, showed pH 7.42, PCO_2_ 49 mmHg, P0_2_: 35.8%, O_2_ saturation 69.9%, HCO_3_ 27.3 mmol/L, base excess 4.6, and lactate 2.3 mmol/L. Abdominal ultrasound revealed moderate hepatosplenomegaly. Whole-exome sequencing (WES) confirmed a homozygous nonsense pathogenic variant in *NPC2* (NM_006432.3: c.141C>A), p. (Cys47*), which was previously prescribed as disease-causing by Chikh et al., 2005 (PMID: 15937921). Additionally, the biomarker lyso-SM509 measured by liquid chromatography—mass spectrometry was significantly elevated: 1.6 ng/mL (≤0.9 ng/mL), in the variant coordinates. 

### 3.3. Others

The bronchoalveolar lavage (Figure 3) was turbid and slightly frothy, and the sedimentation materials in the lower part of the container were suggestive of alveolar proteinosis. They showed numerous macrophages, and the periodic acid-Schiff staining was positive, while the Gram staining and culture were negative. Unfortunately, lipid-laden macrophage, triglyceride, and cholesterol tests were not performed.

### 3.4. Management

Upon admission to our hospital, the working diagnosis was the exacerbation of the interstitial lung disease of childhood (ChILD) with possible lipoid pneumonia based on the clear history of the instillation of olive oil in the nostrils for several months. However, the severity of the lung involvement together with the presence of neonatal jaundice and hepatosplenomegaly raised the possibility of lipid storage disease (Niemann–Pick or Gaucher disease). Awaiting the result of the WES, the patient received supportive therapy, including noninvasive ventilation, a course of intravenous Tazocin, proton pump inhibitor, and hydration therapy with salbutamol, hypertonic saline, and frequent chest physiotherapy. No clinical improvement was observed, and he remained markedly hypoxic despite two weeks of supportive therapy. Thereafter, therapeutic lung lavage was performed. Unfortunately, this procedure was not tolerated well, with a worsening of his respiratory status that required high-frequency oscillatory ventilation (HFOV) for ten more days and then he was switched to conventional mechanical ventilation and remained on this ventilatory mode. He received low-fat feed with a medium-chain triglyceride (MCT) supplement given through a nasogastric tube. As the therapeutic options were limited and the prognosis was poor, the family received genetic counseling and a do not resuscitate code was discussed, unfortunately the patient passed away.

## 4. Discussion

Niemann–Pick disease type C2 is a rare lipid storage disorder characterized by a slowly progressive course leading to death in infancy. Its main clinical features include cholestasis, hepatosplenomegaly, and pulmonary and neurological insult (9, 10). Similar to our patient who presented with cholestasis, Vanier et al. described a child with NPC2 who presented with prolonged jaundice and cholestasis, which was misdiagnosed as galactosemia, and other cases in the literature were misdiagnosed with neonatal haemochromatosis [3]. In our case, presented with jaundice, visceromegaly and early onset of neurological manifestations in the late infantile period, however, the respiratory manifestations were worse compared to published literatures [3,4,7,8]. A young child with unexplained cholestasis, especially if associated with hepatosplenomegaly and hypotonia, should alert physicians to the possibility of NPD.

Niemann–Pick disease type C2 is predominantly respiratory, in contrast to NPC1 [7,8]. There are three types of pulmonary alveolar proteinosis (PAP): autoimmune (the most common), hereditary, and secondary PAP, and coexistence between NPD and PAP is well established in the literature [9,10]. The coexistence of NPD and PAP is the main presentation of NPC2, leading to respiratory failure [11,12]. Similarly, our patient presented with progressive respiratory insufficiency. Furthermore, the instillation of olive oil in his nostrils with possible lipoid pneumonia possibly worsened the respiratory status of our patient [13]. Moreover, the chest X-ray showed homogenous opacity involving the entire right lung and the upper and middle zones of the left lung with an air bronchogram, and the chest CT showed a ground glass appearance together with intrapulmonary fat infiltrate with a Hounsfield unit score of −40 HFU (−40 to −130) PAP [14,15]. Our case showed ground glass opacities, mild smooth interlobular septal thickening, and intralobular lines as a crazy-paving pattern that could suggest a diagnosis of NPD with no evidence of emphysematous lobe, which has been reported as a sign of NPC2 [16,17].

Therapeutic lung lavage is the main supportive therapy in both PAP and lipoid pneumonia, and overlap between the two conditions is common. However, the coexistence of the two conditions has not been reported in the literature to the best of our knowledge. In our experience, gross inspection of the lavage container may help in differentiating between these two conditions. The proteinous material of PAP is likely to sink to the bottom of the container; the color of the lavage is usually pink with a frothy floating substance, as the lipid part will float on the top of the container due to being less dense than normal saline (Table 1). 

No curative therapy is available for NPC2. Multidisciplinary team input would help to support the patient and the family (9, 10). Miglustat is the first and the only approved medication for NPC with neurological manifestations. It is a glucosylceramide synthase inhibitor and an essential enzyme for the synthesis of glycosphingolipids. It has been shown to increase the survival of patients with NPD by five years from the date of diagnosis or approximately ten years from the onset of neurologic manifestations [9,18,19]. There are limited data on other possible therapeutic options, such as stem cell transplantation and liver and lung transplantation. In the current era of genetics, gene therapy may well be an option in the future [20,21,22,23,24,25]. Preventive measures, including antenatal diagnosis and termination of pregnancy with an affected fetus, are viable options for families with a history of NPC2. Additionally, preimplantation genetic testing is the preferred measure if available [9].

## 5. Conclusions

Niemann–Pick disease type C2 is a lethal, progressive, and chronically debilitating disease with multisystem involvement. It starts with neonatal jaundice, with variable degrees of visceromegaly and psychomotor retardation and pulmonary involvement. In this case, NPC2 was misdiagnosed as ChILD, resulting in unnecessary treatment with steroids and hydroxychloroquine, thus highlighting the importance of a detailed history and a vigilant investigation of past illnesses; in this case, discovering the neonatal hepatic manifestations, combined with respiratory and neurological involvement suggested the correct diagnosis. Lipoid pneumonia should be considered as an important differential diagnosis, especially in higher incidence areas such as Saudi Arabia. Lavage is very beneficial to differentiate alveolar proteinosis from lipoid pneumonia. Whole-exome sequencing is the preferred confirmatory diagnostic method. Currently, there is no cure for NPC2.

## Figures and Tables

**Figure 1 children-09-01811-f001:**
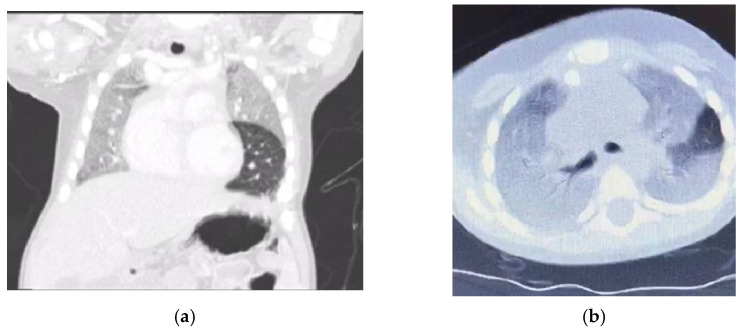
(**a**) CT chest, coronal cut, and lung window with homogenous opacity sparing of lingula but no feature of air trapping or attenuated blood vessels. (**b**) Computed tomography scan chest axial view; lung window ground glass opacity and septal thickening could suggest crazy paving.

**Figure 2 children-09-01811-f002:**
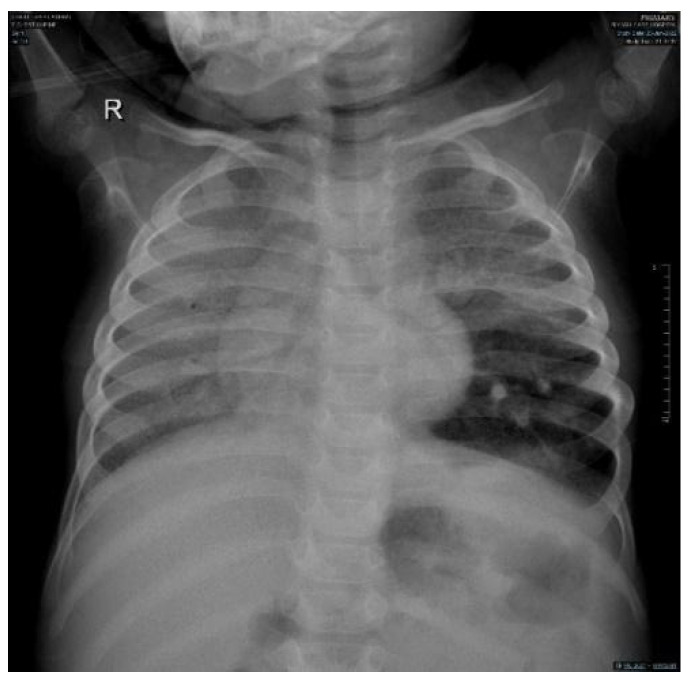
Chest X-ray anteroposterior film shows a diffuse right lung and left upper lobe ground glass opacity with retrocardiac collapse and hyperinflation of the left lower lobe but no oligemia. The mesocardium is explained by the mild rotation to the right side.

**Figure 3 children-09-01811-f003:**
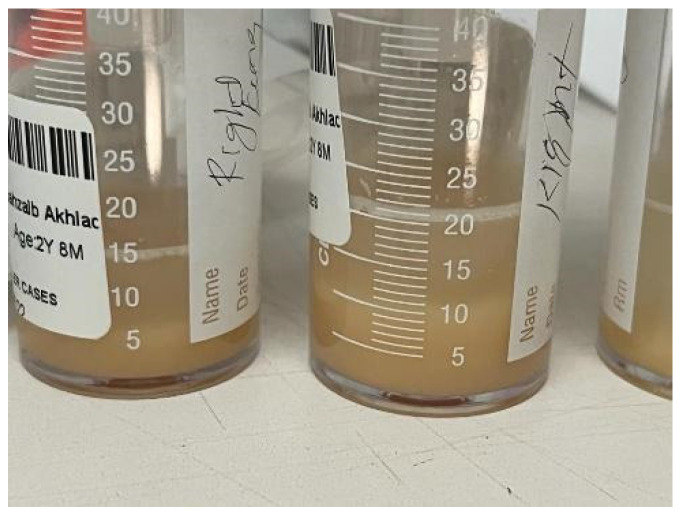
Lung lavage, turbid secretion a little frothy, and sedimentation of material deposited in the lower part of the container suggestive of PAP.

**Table 1 children-09-01811-t001:** Characteristics of Lipoid pneumonia versus pulmonary alveolar proteinosis.

	Lipoid Pneumonia 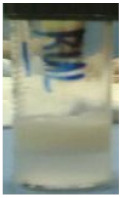	PAP 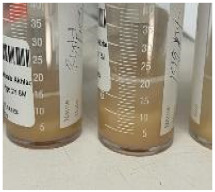
History of lipid ingestion	Yes	No
Deposit in BAL	Floating on the top of the container	Bottom of the container
Cause	Exogenous cause	Autoimmune, genetic, and secondary
Association	GERD	Metabolic diseases
Staining - PAS - Red oil	Negative positive 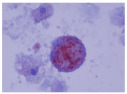	Positive May be positive
Chest X-ray	Ground glass appearance	Bat wing
CT	Consolidation, low attenuation, possible crazy paving	Ground glass, septal thickening, crazy paving
Mycobacterium	Might be possible	Might be positive
Surfactant level analysis	Normal	Might be decreased
Biopsy	Non-eosinophilic material Granuloma, fibrosis	Accumulation of amorphous eosinophilic materials, minimal fibrosis
Treatment	- Discontinue the causative agent - Therapeutic lavage	- Suppurative - Therapeutic lung lavage - GM-CSF - Gene therapy
Prognosis	Good	Variable

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
