# Peer review of "A Niemann–Pick Disease Type C2 with Severe Pulmonary Involvement and Limited Therapeutic Options: A Case Report"

_children, 2022, doi:10.3390/children9121811_

Round 1

Reviewer 1 Report

This case report is focused on the presentation of lung disease in a child with NP-C2, highlighting possible worsening of lung disease due to lipoid pneumonia. The findings are described (see also notes below) and discussed in detail. The article is interesting and should be published, the manuscript, however, requires re-reading and corrections of typos.

Questions:

1) Figures 1 and 2 are - at least in the pdf. file provided to me - of low resolution, on which it is difficult to distinguish details. The images should be printed larger and have a higher resolution. More detailed interpretation and description of radiological findings would be helpful - please note that this reviewer is neither a radiologist nor a pulmonary specialist:

Figure 1(b) : is the appearance of changes with consistent with "crazy-pavement" pattern as described by von Ranke (Figure 3) - the mix of ground-glass opacities and interlobular thickening 1 ? Figure 1, 2: Can the "lung window with low attenuation" be regarded e.g. as emphysema ? Elleder et al.2 described patchy areas of consolidation mixed with areas with increased aeration as characteristic in NP-C2.

2) BAL in NPC typically contain storage macrophages, which may (and should) lead to diagnosis. What was histological appearance of macropages in BAL? Was it consistent with lysosomal storage ? Was it suggestive of lipoid pneumonia ? A microscopic picture of macrophages from BAL should definitely be included in the article, if available.

3. Was the mutation novel ? A brief comment on the mutation and its presumed severity should be included - a premature stop codon. I am not sure what means "class 1 pathogenic variant" in the context of NP-C2, other description would perhaps be more helpful.

As an aside which goes beyond the scope of the paper it is interesting to note that hydroxychloroquine with which the patient treated in the past, is a lysosomotropic drug.

A partial list of typos and changes needing corrections - this is not exhaustive list of typos, please re-check the manuscript

Title - correct "Neimann Pick! to Niemann-Pick

Abstract
"His parents used to install olive oil in his nose" instill ?
dtto - page 4, line 117: "history of installation of olive oil in the nostrils for several months." instillation ?

Page 5 line 169 : "Migalastat is the first and the only approved ..." - the drug approved for treatment of NP-C is Miglustat, not Migalastat. ...

1. von Ranke FM, Pereira Freitas HM, Mançano AD, et al. Pulmonary Involvement in Niemann–Pick Disease: A State-of-the-Art Review. Lung. 2016;194(4):511-518.

2. Elleder M, Houštková H, Zeman J, Ledvinová J, Poupětová H. Pulmonary storage with emphysema as a sign of Niemann–Pick type C2 disease (second complementation group). Report of a case. Virchows Arch. 2001;439(2):206-211.

Author Response

What amazing feed back 

Many thanks 

 Very helpful really 

 Dr. Shamrani 

Reviewer 2 Report

Dear Authors,

this is a very interesting paper about a very rare disorder. The experience in this case is very important. The title reports a wrong name of the disease Niemann instead Neiman.

I suggest to delete in the abstract the word "fatal" because NPC has some therapies and also late onset mild form. The phrase about the drops in the nose is not pertinent in the abstract, you can reserve it inthe discussion. 

The case report is well writtend and detailed but for pulmonary insufficiency you need to do the arterial blood gas. Which was the lipid assessment at blood examination? The patient showed dysphagia?

Could you please elaborate bettere the conclusions with differential diagnosis? 

Author Response

Great in put 

Deeply appreciated 

 Many thanks 

see the feedback please

DR SHAMRANI

Reviewer 3 Report

This is a nice report on an additional, rather typical case of Niemann-Pick disease type C2, whose lung involvement was aggravated by protracted nasal administration of olive oil thought to be beneficial for the patient‘s epistaxis. The authors impressively describe the characterization and management of the lung disease in this patient where PAP as a sign known for NPC2 but also pulmonal lipoidosis possibly caused exogenously were present. They mention the differential diagnosis of childhood interstitial lung disease but underline that in Niemann-Pick disease types such a condition must not be (but here in the initial absence of the NPC2 diagnosis was) treated with steroids and hydroxychloroquine. Thus, the study contributes to a better understanding and management of the lung disease in Niemann-Pick disease types.

However, within the spectrum of described NPC2 patients, the study provides no definitely novel aspects except for the probably new homozygous nonsense variant in the NPC2 gene. 

I have the following points:

— Please correct in the title: Neimann Pick to Niemann-Pick

— Please complete at the beginning of Results section 3.1. in line 87 the first sentence by writing about: There were homogenous ground glass opacities involving ...

— Please correct in lines 93/94 (Results section) homogenous to homozygous. Moreover, it seems inappropriate to include the molecular and biochemical data (lines 93 to 96) in the Results subsection 3.1. entitled Chest X-ray (Figure 2). I would suggest to introduce a small subsection 3.3. of Results, entitled Confirmatory molecular and biochemical findings; or something like this, with the data of lines 93 to 96. It appears also to be necessary to indicate, what laboratories or which of the present co-authors have provided the data. This may be done as Acknowledgements or in the section Author Contributions at the end of the manuscript. That data are pivotal to the study, thus should be somewhat highlighted and authorized. Please indicate also whether the reported nonsense variant of the NPC2 gene is novel (as I think) or not.

— Given that the severe lung involvement in NPC2 has been known since the late 1990ies to be relatively characteristic of this disease type,  some of the old literature could be cited. I recall, for example, Eur J Pediatr. 1998 Jan; 157(1):45-9, but others are also possible. When dealing with lung involvement in that condition, some historical aspect may be nice. 

— Discussion, line 169: You erroneously write Migalastat. The name of the drug for NPC1/2 is Miglustat, i.e., the glucose derivative, not the galactose derivative.

Author Response

Great in put

many thanks 

 deeply appreciated 

 dr shamrani
